# Anti-COVID-19 Credentials of Chitosan Composites and Derivatives: Future Scope?

**DOI:** 10.3390/antibiotics12040665

**Published:** 2023-03-28

**Authors:** Judy Gopal, Manikandan Muthu, Suraj Shiv Charan Pushparaj, Iyyakkannu Sivanesan

**Affiliations:** 1Department of Research and Innovation, Saveetha School of Engineering, Saveetha Institute of Medical and Technical Sciences (SIMATS), Thandalam, Chennai 602105, India; 2Department of Bioresources and Food Science, Institute of Natural Science and Agriculture, Konkuk University, 1 Hwayang-dong, Gwangjin-gu, Seoul 05029, Republic of Korea

**Keywords:** chitosan derivatives, composites, antiviral, COVID-19, antimicrobial, natural compounds

## Abstract

Chitosan derivatives and composites are the next generation polymers for biomedical applications. With their humble origins from the second most abundant naturally available polymer chitin, chitosan is currently one of the most promising polymer systems, with wide biological applications. This current review gives a bird’s eye view of the antimicrobial applications of chitosan composites and derivatives. The antiviral activity and the mechanisms behind the inhibitory activity of these components have been reviewed. Specifically, the anti-COVID-19 aspects of chitosan composites and their derivatives have been compiled from the existing scattered reports and presented. Defeating COVID-19 is the battle of this century, and the chitosan derivative-based combat strategies naturally become very attractive. The challenges ahead and future recommendations have been addressed.

## 1. Introduction

The interest in natural and man-made polymers for their use in biomedicine and related applications has seen a steady rise in recent years. Numerous polymers, irrespective of their biodegradability, have been examined in this respect. Some of the most preferred synthetic polymers were polylactic acid or PLA, polylactic-co-glycolic acid or PLGA, poly-caprolactone or PCL, acrylic polymers, and polyethylene glycol or PEG, whereas the most researched natural polymers were chitosan, gelatin, dextran, cellulose, starch, alginate, and hyaluronic acid [1,2,3]. The naturally occurring polysaccharide, chitin, was first extracted by French scientist Henri Braconnot from mushrooms and, thereby, laid the foundation for carbohydrate-based polymers [4]. Chitin was obtained from insects in the 1830s, before its successful transformation to chitosan in the late 1850s [4]. Deacetylation (removal of CH_3_-CO group) transforms chitin to chitosan [5], a special cationic polymer that is soluble in weak acids. The level of deacetylation determines how soluble chitosan will be. More than 80–85% of the product must be deacetylated in order to produce a soluble product [6,7,8].

Although chitosan is regarded to be eco-friendly and generally regarded as a benign and non-irritating substance, the FDA has not yet approved its usage in pharmaceutical compounds owing to a number of restrictions [9]. In the USA and European and Asian countries such as Finland, Italy, and Japan, it is legally in use as an obesity control supplement. The chitosan monograph was added to the US National Formulary in 2008 and the European Pharmacopoeia in 2011 [10]. It is currently recognized as a secure (GRAS) excipient for treating wounds. In addition, the FDA has approved a number of non-categorized chitosan devices for blood clotting and wound dressing [11]. Among them are Axiostat^®^, Celox™, and ChitoFlex^®^, chitosan-based hemostatic bandages (currently marketed in numerous countries) that hasten blood clotting following an injury or trauma. Another brand named TraumaStat^®^ is a fabric-based dressing that helps surgical wound reparation, with ease to paste and remove at will [12]. Some products such as Anscare, which is a chitosan bandage in Taiwan, is available mainly to treat external wounds. In addition to having the property of initiating faster blood clots, chitosan possesses numerous biomedical applications: antibacterial [13,14,15], gene delivery [16], bio-composites [17,18,19,20], nanoengineering [21,22,23], and excipients [8,24]. Structurally chitosan is a positively charged polysaccharide, which makes it easier for it to adhere to the surfaces of the bacterium. By optimizing physical properties such as wettability, the hydrophobicity of chitosan is enhanced, helping it penetrate the lipid bilayer of the bacterium and affecting the integrity of the cell wall. Eventually, lysis in bacteria results due to cytoplasmic leakage [25,26].

Antimicrobial resistance (AMR) to drugs has raised the need for seeking alternative treatment possibilities. This need is further compounded by pragmatic limiting factors such as the difficulty in finding alternative antimicrobial medicines in consideration of emanating factors pertaining to the rapid rise of antimicrobial resistance, alternative antibiotic availability, and side effects of incessant antibiotics usage [27]. Some of the biggest threats to public health are the looming risk of increasing bacterial resistance to antibiotics and the faster spread of its resistant strain, slow progress of a new class of antibiotic development, and abuse of existing antibiotics [28,29]. All these factors act as inhibiting factors leading to therapeutic failures. The investigation of the antipathogenic of natural occurring compounds is significant since synthesis and clinical trials of alternative antibiotics are in high demand [30]. Abundance-wise chitosan is the second best available natural polymer, and it has already proven its utility in the medical, pharmaceutical, food industrial, and techno-textiles industries due to its exceptional properties of bio-friendliness (in terms of compatibility and toxicity), combined with eliciting other favorable physiological properties such as antioxidant, anti-cancerous, and immune-modulatory functions [31,32].

One of the barriers to the therapeutic application of chitosan is its limited solubility at body or physiological pH [33,34]. Various strategies have been sought after, in order to optimize the chitosan chemistry and to resolve the low solubility issue. Typically, chemical alterations in the structure are done at three active sites, including one at the main amine (-NH2-) group and two hydroxyl groups (-OH) on the glucosamine unit. Based on attached chemical moieties, the characteristics of chitosan have been manipulated. However, the modification in the main amine is crucial for successful transfection through the cell membrane. It has been observed that chitosan’s chemical alteration enhances its medical applications and pharmacological capabilities. For instance, p-azidebenzoic acid addition at the main amine of chitosan increases cohesiveness to the cell membrane, thereby enhancing the rate of healing abilities [35], whereas bactericidal potential of chitosan is increased through PEGylation [35,36]. For the respiratory administration of siRNA, piperazine-substituted chitosan that is soluble in polar solvents is utilized [37]. Moreover, the addition of methyl methacrylate moiety within chitosan has shown to be useful for drug delivery application [38]. Similarly, N-carboxymethyl chitosan has wider applications in biomedicine owing to its enhanced binding and water-soluble properties [4,39,40]. Several studies have examined the potency and biosafety of chitosan for use in applications, ranging from orthopedic tissue engineering [41,42,43] to targeted delivery [16,44], biomaterials [17,45], and medicinal excipients [24,32,46]. Equally there are reviews focusing on toxicity associated with chitosan [10] and chemically-modified chitosan [38,47,48,49,50].

COVID-19 or SARS-CoV belongs to the family of Coronaviridae that consists of single-stranded RNA viruses. Its symptoms include gastrointestinal, pulmonary, hepatological, and neurological symptoms, the effects of which span from moderate to lethal. The size of the COVID-19 virus measures roughly 60–160 nm in diameter including the periphery under an electron microscopy. An envelope of coronavirus consists of a structure containing single-stranded RNA (+ssRNA) with a genome size of 27–32 kb, with a 5’-cap structure and a 3’-poly A tail that interacts with the nucleoprotein. As a result, when the virus enters the recipient, it invades the cells by binding onto the ACE2 (angiotensin-converting enzyme) receptor on the cell’s surface through its surface-bound glycoprotein component (4–10). Consequently, after infection the RNA of virus is transferred into the host cell cytoplasm and begins replication involving the translation of two structural polyproteins. Replicated glycoprotein envelopes are stored in the Golgi complex or endoplasmic reticulum membrane. Finally, the nucleocapsid is created by combining genomic RNA and glycoprotein. Inside the Golgi complex, germination of viral particles takes place. Ultimately, the plasma membrane and the compartments containing the viral particles merge to discharge the virus. Currently, reverse transcription polymerase chain reaction (RT-PCR) kits and CT (computed tomography) scans taken at various angles in pulmonary area are being dominantly used to diagnose SARS-CoV-2 positive patients [51]. The drugs currently in use to treat COVID-19 are Remdesivir, Enoxaparin, Methyl Prednisolone, Dexamethasone, Tocilizumab, and Ivermectin (https://www.fda.gov/drugs/emergency-preparedness-drugs/coronavirus-covid-19-drugs (accessed on 4 February 2023).

In the following sections we present an overall perspective on the antiviral applications of chitosan and its derivatives. Chitosan composites and derivatives have been compiled and presented for the first time, specifically for anti-COVID-19 applications. The difficulties faced, its limitations, and suggestions for its future course have been presented. The obstacles facing the use of chitosan and the recommendations for its future have been briefly discussed in this review.

## 2. Snapshot of Antipathogenic Applications of Chitosan Composites and Its Derivatives

Chitosan is one of the compounds that has been predominantly studied for their various medicinal and biological functions. It has also been investigated as a potent antimicrobial agent that is effective against a large variety of microbes such as bacteria, viruses, and fungus [52]. In this section, we will present a snapshot of the specific antimicrobial applications of chitosan composites and derivatives. Chitosan has been demonstrated to exhibit fungicidal properties against *Candida* strains, which are responsible for a wide range of ailments in mammals, animals, and bacterial species. Similarly, chitosan has exhibited bactericidal activities against *Staphylococcus aureus* (gram-positive), *E. coli* (gram-negative) [53], and strains such as *Bacillus megaterium*, *Salmonella typhimurium*, and *Pseudomonas fluorescens*, to list a few [54]. Chitosan-bound linens are being tested in order to enhance quality of life by inhibiting *Staphylococcus aureus* growth when in contact with human skin. Various clinical testing is underway [55], with human trials [56]. A study is now being conducted on 165 subjects for testing the therapeutic potency of chitosan-based textile (DermaCura^®^, a brand that contains about 1% chitosan matrix) [56]. Apart from textiles, chitosan nanoparticle-loaded VESTA respirators were developed against pathogens. A clinical study (NCT04490200) (https://clinicaltrials.gov/ct2/show/NCT04490200 (accessed on 10 February 2023)) involving N95 masks with and without chitosan is underway on 700 clinical professionals to check for its effectiveness against COVID-19 infection, and whether the chitosan nanoparticles can adsorb and inactivate the virus with which come into contact. At Jinan Military General Hospital, 120 patients were enrolled in a clinical trial for anti-microbial chitosan dressing gel for the treatment of nonneoplastic epithelial diseases involving the skin and vulvar mucosa (NCT02890277) [57].

Right now, some of the notorious bacterial pathogens resisting antibiotic treatments are vancomycin-resistant *S. aureus* (VRSA), methicillin-resistant *S. aureus* (MRSA), methicillin-resistant *S. epidermidis* (MRSE), *E. coli*, vancomycin-resistant Enterococcus (VRE), *Acinetobacter* spp., *Pseudomonas aeruginosa*, and *Klebsiella pneumoniae*. Park et al. looked into the antibacterial effect of chitosan against drug immune *Pseudomonas aeruginosa* and *S. aureus* [58]. The findings demonstrated that all isolates of *P. aeruginosa* that were drug resistant, as well as some variants of *S. aureus*, were more susceptible to chitosan’s antibacterial activity [58]. Furthermore, an in vivo study of chitosan on mice that were infected by bacteria revealed a better survival rate and minimal bacterial colonization [58]. In another study, similar results of chitosan’s antibacterial potential against ATCC strains and *E. faecalis* have been noted [59]. In a cross comparison of available strains and clinical multi-resistant extracts, Costa et al. assessed the antibacterial effect of chitosan towards MSSA (methicillin-susceptible *Staphylococcus aureus*) and MRSA (methicillin-resistant *Staphylococcus aureus*). The findings demonstrated that methicillin resistance had no effect on chitosan’s activity. Additionally, MRSA seems more susceptible especially with a decrease in the molecular weight of chitosan [60]. Given the clinical significance and current treatment scenario for MRSA infections, these results are very positive. *Acinetobacter* spp. infections too pose grave concern as they have the extraordinary capacity to build resistance and spread in the hospital setting [61]. From this perspective, chitosan offers respite naturally, which makes its offer more attractive and appealing. In addition to chitosan itself, its derivatives and allied composites have also demonstrated their antimicrobial activity. In an attempt to combat *P. aeruginosa*, MRSA, and *A. baumannii*, Saito et al. produced chitosan conjugated with lysozyme-oligosaccharides to exert strong antibacterial activity [61]. Interestingly, the efficacy of chitosan nanofibers against multiple drug-resistant strains of *Clostridium difficile* was evaluated and found to be effective [62]. Zhang et al. [63] fabricated a chitosan nanoparticle encapsulated with antibacterial oils and tested its efficacy against multiple drug-resistant *K. pneumoniae*. The results suggested that chitosan with a lower molecular weight and deacetylation level of 75% tends to boost the bactericidal effects of oils [63]. Loading chitosan nanoparticles with drugs like cephalosporin, cefotaxime [64], and ceftriaxone for their application against isolates—including a wide range of multiple drug-resistant strains such as carbapenemase producing *K. pneumoniae*, *E. coli*, *P. aeruginosa*, and MRSA strains such as *E. coli* and *P. aeruginosa* [64]—has produced desirable antimicrobial properties. The findings showed significant improvement in antibacterial activity across various bacterial strains. Considering the current scenario of AMR, the fabrication of various novel chitosan derivatives-based drugs has become an extended application.

Chitosan of low molecular weight of about 70 kDa and an acetyl removal degree of about 75% showed promising antifungal efficacy [65]. A 15 kD chitosan-oligosaccharide with powerful synergistic actions against *Candida* spp. was described by Ganan et al., when used in conjunction with several antifungal medications such fluconazole, miconazole, voriconazole, and amphotericin B [66]. Lo et al. investigated the antifungal efficacy of chitosan by modifying the molecular weight and deacetylation degree in conjunction with drugs like caspofungin, fluconazole, and amphotericin B. A combination of chitosan and fluconazole produced impressive synergistic antifungal effects, while other antifungal drugs had no effect. In addition, chitosan and fluconazole worked well to treat drug-resistant strains of *Candida* species albicans and tropicalis strains [67]. The alarming incidence of fungal infections in recent times at a global scale is a serious concern, especially given their effect on vulnerable populations who already suffer from various diseases or disorders such as diabetes and are immunocompromised. Fungal infection, be it on its own or after initial bacterial/viral infections, creates a cascade of related issues, which worsens the outcome of treatment protocols [68,69]. The pathology reports of documented human deaths suggest over 90% lethality due to fungal infections attributed to the genus *Candida*, *Pneumocystis*, *Aspergillus*, and *Cryptococcus* [68]. According to the evidence in possession, fungi seem to be more vulnerable to chitosan’s antipathogenic activity than bacteria [14]. Chitosan’s fungicidal activity can be manipulated by varying factors like molecular weight, content or volume, formulation pH, and acetyl group removal level [67]. Based on the strain, factors such as deacetylation levels and their molecular weights vary and are able to affect its fungicidal effects [67]. Normally, chitosan concentrations between 1% and 5% were shown to guarantee optimum antifungal activity, especially at low pH [14,65]. The mechanism of chitosan’s fungicidal activity initiates rupturing of the fungal cell wall after its transfection across the yeast cell [70]. Chitosan antifungal effect against *Candida* spp. initiates through uptake of Ca^2+^, with efflux of K^+^ along with inhibition of respiration and other cell activity such as fermentation [70]. It must be noted that *Candida* species are the most frequently studied pathogenic yeast with respect to the fungicidal action of chitosan. The influence of the molecular weight of chitosan on 105 clinical *Candida* isolates, including their strains that were resistant to fluconazole, has been investigated. Results showed that chitosan had substantial antifungal effect, blocking >89.9% of the isolated strains. Additionally, chitosan demonstrated a similar antifungal effect at lower concentrations than fluconazole when evaluated on nine investigated patients [65]. However, the outcome of chitosan investigations in synergy with antifungal medications is contradictory. However, there is no doubt that it does possess strong antimicrobial attributes. Antimicrobial drugs have been delivered by controlled release mechanisms using chitin and its derivatives [71]. Pharmaceutical technology makes considerable use of chitin derivatives, including N-succinyl-chitosan, carboxymethyl-chitin, and hydroxyethyl-chitin [72]. More such systems have been developed by encapsulating several active principles with chitosan, including proteins and peptides, growth factors, anti-inflammatory chemicals, antibiotics, and molecules for anticancer therapy as well [73]. Thus, a lot has been achieved in the field of antimicrobial activity through chitosan and its derivates/composites.

## 3. Antiviral Applications of Chitosan Composites and Derivatives

The antiviral action of chitosan and its derivatives is less understood and explored than their antibacterial and anticancer activities. This is most likely due to the challenges and particular requirements for virus cultivation. We summarize the antiviral activity of chitosan and its derivatives, and composites, in this section. Figure 1 and Figure 2 show the chemical structures of the known chitosan derivatives that have shown promising results against viruses. In a mouse model, Zheng et al. found that intranasal delivery of chitosan is efficient in preventing influenza A H7N9 infection, with the proposed mechanism being the stimulation of the innate immune system [74]. Chitosan has recently been suggested as a potential chemical agent against the SARS-CoV-2 virus by Sharma et al. [75]. The hypothesized mechanism is linked to targeting CD 147 receptors, an unique pathway for virus entry into cells [75,76]. Studies show that chitosan may operate as a drug carrier, and that it can be utilized to develop targeted delivery systems that can improve the therapeutic efficacy of some antiviral medicines such as foscarnet [77], acyclovir [78], ribavirin [79], and amantadine [80]. Several studies have suggested that HIV-1, Sindbis virus, the herpes simplex virus, human cytomegalovirus, the Japanese encephalitis virus, and several other parvoviruses, flaviviruses, picornaviruses, retroviruses, alphaviruses, and papillomaviruses use the heparan sulphate binding receptor (GAGs) as the preliminary pathway for random cellular entry and pathway mediation [81,82,83,84,85,86,87]. Artan et al. discovered that a chitosan-oligosaccharide (3–5 kDa) inhibited HIV-1 replication by suppressing HIV-1-induced syncytia formation and reduced p24 antigen expression [88]. This oligosaccharide may also be able to prevent viral entrance and virus–cell fusion. This chitosan derivative has the potential to be a unique agent for the build-up of a new class of anti-HIV medicines [88]. The work of Artan et al. [88] on the suppression of HIV-1 strains by chitosan derivatives revealed that sulfation of functional amine and hydroxyl groups can boost anti-HIV-1 cues, especially at low molecular weight (MW 3–5 kDa) and concentration. The EC (emulsifiable concentrate) of sulfated chitosan oligosaccharide was 2.19 g/mL, and it had a lytic effect (EC 1.43 g/mL) [89] against HIV-1 virus. Furthermore, studies show that sulfated bound chitosan derivatives prevent infected cell-to-cell merging or syncytia formation, as well as block contact between the HIV-1 strain and host cell receptor CD4+ [88]. Sulfate replacement on the chitosan backbone was also proposed. Another study used MT4 cell lines to investigate the antiretroviral ability of a chitosan-based conjugate and nanoparticles. It led to the conclusion that chitosan-conjugate nanoparticles might be employed for targeting and producing long-lasting polymeric prodrugs with increased therapeutic potential and fewer unwanted effects for antiretroviral therapy [90]. Their effect against hepatitis C virus genotype 4a using human hepatoma cells showed that they completely inhibited viral entry and replication [91]. Figure 1 displays the various chitosan derivatives that have been related to antiviral activity.

A study of the Moloney murine leukemia virus with heparan sulphate receptor investigation on SC-1 and NIH-3T3 cell lines concluded that sulphate replacement on C_4_H_6_O_4_ of succinyl chitosan structure exhibited a reasonable inhibitory effect out of dextran sulphate chitosan matrices as a positive control. The antiviral efficacy was observed to get higher with molar mass and degree of sulphate substitution [92]. Nishimura et al. investigated the addition of sulfonic groups to an amine group of carbon-2 (C-2) and a hydroxyl group of carbon-3 (C-3) vs. carbon-6 (C-6) [93]. In addition, the addition of sulfonic groups to the hydroxyl group of C-6 did not have a significant effect on anti-HIV-1 efficacy. They hypothesized that the anti-HIV-1 activity of sulfonic groups at C-2 and C-3 is due to anticoagulant interactions with the GP120 envelope glycoprotein, whereas replacement at C-6 had non-specific ionic interactions with blood coagulation proteins [93]. This inference of random ionic interaction resulting from sulphate substitution was supported by Yang et al. in their coagulation assay study evaluating the anticoagulant efficacy of sulfated chitosan with sodium heparin as the standard [94]. The activated partial thromboplastin time (APTT) and thrombin time (TT) experiments established a correlation between anticoagulant activity of sulphate substituted chitosan with proportion, molecular weight, and degree of sulphate substitution, indicating chitosan sulphate’s role in modifying intrinsic mechanism of coagulation. In support of previous research [95], the ionic interaction was investigated as a possible explanation for the surface charges on sulfated chitosan and their binding potential with cell receptors. The capacity of N-carboxymethyl chitosan-N-O-sulfate, a sulfated derivative of chitosan, to suppress HIV-1 infection in CD4+ cells was investigated. It prevented virus attachment to CD4+ cell receptors, virus adhesion to host cells, and the competitive suppression of reverse transcriptase enzyme. Its ampholytic nature and high and localized negative charge were related to the mechanism of action. The anionic derivative of chitosan inhibited in a dose-dependent manner without causing considerable damage to the host cell [95].

Jayakumar et al. reviewed the anti-HIV1 activity of chitosan and its derivatives [96]. They noted a mechanism of inhibition of viral multiplication and its binding to the host’s CD4 receptor. They also emphasized site selectivity, citing the improved anti-HIV-1 efficacy of polysaccharides sulfated at O-2 and/or O-3 sites [96]. Dimassi et al. have described a range of physiochemical properties and potential substitution positions of chitosan that suited a spectrum of biomedical applications [97]. The authors underline that chitosan sulfonation at various sites of free amino groups and/or hydroxyl gives excellent antibacterial activity, comparable to heparin or other sulfated glycans that bind via many intracellular and extracellular inhibitory pathways. They also allude to a reported study by Sosa et al. [95,96], that employed the random sulfonic substitution technique to test the HIV-1 reverse transcriptase and CD4 receptor binding inhibitory capabilities along the aforementioned lines. The study also indicates that sulfated chitosan stimulates an immune response via nitric oxide. In order to reduce HIV-1 infectivity, low molecular weight chitosan oligomers (COS) were modified with tryptophan (W), glutamine (Q), and methionine (M) in various proportions. The results revealed significant inhibition of syncytia formation, cytopathic effect, and viral replication, as well as a downregulation of levels of p24 antigenic protein visible on virus-affected cells, envelope protein, and viral infection factor proteins, indicating a potent antiviral activity against multiple HIV-1 strains. By altering the binding procedure, QMW-COS blocked the contact between the host cell and virus. However, WMQ-COS had a smaller effect [98]. When loaded with tenofovir, polyelectrolyte nano-complexes of chitosan and hyaluronan stabilized with zinc (II) showed synergistic effects. Internalized into human peripheral blood mononuclear cells infected with HIV-1, these nanocarriers greatly reduced viral infection by blocking the formation of capsid p24 protein [99].

Ishihara et al. evaluated the effectiveness of altered chitosan component against influenza virus A, resulting in intriguing findings [100]. Mori et al. employed chitosan silver nanoparticle composites as a nanofiber sheet to capture and immobilize virions for enhanced virucidal effects of silver nanoparticles against influenza virus A [101]. In addition, Cheng et al. created chitosan sialyl oligosaccharides complexes and discovered that this complex is able to inhibit influenza virus glycoprotein’s contact with cell receptors for viral entrance [102]. Collin, Paulson, et al. and Lee et al. [103,104] argued that glycoprotein cluster production impairs viral affinity to the cell surface to explain the inhibitory mechanism. Chitosan suppressed influenza virus infectivity when injected intra nasally in a mouse model against a rapidly evolving strain of new avian influenza A virus: H7N9 virus, a causative agents of several respiratory tract disorders. The investigation demonstrated a dosage-dependent protective effect and a proportional survival rate, emphasizing sulfated chitosan’s antiviral efficacy against HIV-1. Both the attachment of GP120 receptors to T-cell CD4+ receptors and the HIV-1 reverse transcriptase enzyme were inhibited. In contrast, sulfated chitosan oligosaccharide inhibits the syncytium formation of HIV-1-infected CD4+ cells (the merging of infected cells with nearby healthy cells to generate multinucleate large cells). This reduces virus multiplication and infection dissemination. (Survival rates began to diminish when chitosan concentrations decreased from 30 g to 10 g) They also found a reduction in viral load, a rise in pro-inflammatory signaling molecules, and leukocyte invasion in lung/trachea tissues. Impressively, the stimulated immune behavior remained for 10 days. Nonetheless, even at the greatest dose of chitosan (100 g), only 10% of the animals were protected when the route was changed to intraperitoneal. Compared to H1N1 and H9N2 strains, the PR8 strain had only limited protective effect [74]. Moreover, in the presence of chitosan, nickel cations displayed the maximum attachment for enterovirus-71 because Ni2^+^ includes six accessible coordination links. Chitosan chelated with nickel ions improved Ni2^+^ chelation with enterovirus-71 protein and bond formation stability [105]. Pauls discovered the antiviral effect of chitosan on an adenovirus that causes pulmonary issues in young children [106]. Flow cytometry was used to confirm that the 0.1% chitosan-treated NIH-3 T3 cell lines of mouse embryo tissue infected with green fluorescent protein (GFP)-adenovirus displayed the highest drop in fluorescence compared to the positive control group. Moreover, 0.1% chitosan exhibited the best cell activity compared to 0.5% and 1% [106]. The human papillomavirus (HPV) has been identified as a leading cause of sexually transmitted diseases (STD). Since vaccines are prohibitively expensive, and vulnerable groups in lower-income countries are at a higher risk, researchers are constantly investigating innovative approaches to resist viral infection and its spread. Given the demonstrated antiviral activity of sulfonated chitosan, Gao et al. studied the replacement of hydrogen ions at carbon 3 and 6 of chitosan with sulphate groups, and found intriguing results [107]. The modified chitosan, 3,6-O-sulfated chitosan (58.3 kDa, 45.8% sulphate), suppressed HPV pseudoviruses (HPV PV). The suppression was dosage-dependent, and chitosan interacted with HPV PV capsid with a high selectivity index (CC50/IC50 = 1222.3), preferably at low pH conditions (pH = 5). Other investigations on the HeLa cell line demonstrated that sulfated chitosan requires a lengthy incubation period to enter inside host cells, whereas host cell-cell surface adhesion happens almost instantly. Using western blot analysis, it was found that the activation of L1 capsid protein was drastically decreased in HPV-infected cells, indicating that sulfated chitosan suppressed the cellular PI3K/Akt/mTOR pathway [107]. Sulfate substitution has conferred an advantage on derivatives over other compounds. Ishihara et al. [108] investigated sulphate group addition for carboxyl methyl functional group in chitin structure. At high doses (100 and 300 g/mL), it reduced Friend murine leukemia virus foci in dunni cells and the herpes simplex type-1 virus. Similar to dextran sulphate, the decrease displayed dose-dependence when delivered during viral absorption. Before or after viral uptake, Vero and ATCC-VR-539 cell lines exhibited minimal activity in vitro [108]. On a Vero cell line, the antiviral activity of chitosan nanoparticles derived from Lucilia cuprina maggots was evaluated for following viruses: herpes simplex virus (HSV-1-a DNA virus), coxsackie viruses (RNA virus) and Rift Valley fever virus (RVFV). When compared to the control, the degree of viral suppression noted to be high for RVFV (24.9%) followed by coxsackie virus (26.1%) and HSV-1 (18.8%), post treatment with chitosan nanoparticles [109]. Graphene is used as a protein absorbent to eliminate viruses in drinking water in combination with N-2-hydroxypropyl-3-trimethylammonium chitosan chloride (HTCC) [110]. Mi et al. [111] created an electro-spun mat using polyvinyl alcohol (PVA) monomer crosslinked with glutaraldehyde to reduce the hydrophilicity of HTCC polymer. Such modification of surface property enables quaternized chitosan derivatives for its use in water treatment to selectively absorb viruses. In fact, studies reported a higher log elimination value for Sindbis virus (enveloped) and Porcine parvovirus (non-enveloped) [75,111].

HTCC (250 kDa) inhibits HCoV-NL63, HCoV 229E, HCoV OC43, and HCoV HKU1 [112,113]. As a function of medium ionic strength, HTCC and HTCC nanospheres/microspheres electrostatically attach to the recombinant ectodomain of the coronavirus S protein, thereby creating the matrices made up of protein–polymer complexes [112,114]. This stops viruses like HCoV-NL63 from attaching to ACE2, inactivating them. Sulfated chitin is less hydrophilic than natural chitin. Sulfated chitin with low molar mass (16–58 kDa) possesses antiretroviral activity for HIV [93]. The position of sulfation influences anti-HIV efficacy more than the overall amount of sulfated chitin substitution [93]. Less sulfate-substituted 3S sulfated chitin is more efficient against HIV-1 than 6S sulfated chitin. Sulfation of the O-6 N-acetylglucosamine residue reduces chitin’s anti-HIV-1 efficacy. The chitin sulphate moieties at O-3 and/or O-2 are hypothesized to generate a unique interaction with HIV-1 gp120 [93]. The electrostatic attraction between negatively charged HIV gp120 to positively charged T lymphocyte receptors [115] cause HIV to enter host T lymphocyte cells. With the conjugation of HIV gp120 with 2,3-Osulfated chitin [116], infection of T cells by HIV can be avoided. Typically, sulfated chitosan and oligochitosan with molecular weights of 1.277–1000 kDa, 0–25% acetylation, and 0.82–1.55-degree sulphate substitution exhibited antiviral activity against HPV, HIV-1, VSV, and NDV [117]. The virus–cell fusion within the host cell is prevented due to bonding of glycoprotein (gp) receptors or viral capsid proteins with chitosan derivatives [107,117,118,119]. Alternately, they inhibit autophagy, which is necessary for viral entry, by downregulating host cell biological pathways, such as the PI3K/Akt/mTOR pathway [107]. Chitosan sulfation is essential for providing HIV and other enveloped viruses with effective and selective viral suppression. Chitosan derivatives with higher sulphate content and larger molecular weights (100 kDa) had better antiviral activity compared with lower molar mass (25 kDa) [92]. According to Yang et al. (2018), sulfate substituted chitosan (5.030 kDa) triggers more nitric oxide generation inside hosts, in comparison to higher (14.481 kDa) and smaller (3.169 kDa) counterparts [120,121]. In another investigation, sulfated chitosan oligosaccharides with lower molecular weight of about 1 kDa have hampered the amount of nitric oxide formation [121]. Sulfated 6-O-carboxymethyl-chitin (SCM-chitin) (430 kDa, 0.0766–1.69 degree of sulphate substitution, 0.56 degree of carboxymethyl substitution) greatly suppresses Friend murine leukemia. Viruses can infect host cells by connecting to specific receptors on host cells. Since F-MuLV and herpes simplex virus do not share a common host cell adsorption receptor, it is likely that SCM-chitin exerts its antiviral activity by non-specifically attaching to the viruses. The presence of a sulphate moiety and/or a large molecular weight (430 kDa) in SCM-chitin may be significant factors in its antiviral action, as CM-chitin, which has no structural sulfate with a molecular weight of 63 kDa, has shown no inhibitory impact against FMuLV and herpes simplex virus [108]. It has been demonstrated that N-carboxymethylchitosan N,O-sulfate (NCMCS) inhibits HIV-1 multiplication within human T helper cells and Rausher murine leukemia virus replication [95]. NCMCS has a molecular mass of 7.4 kDa, an acetylation degree of 16%, and an N-carboxymethyl substitution degree of 0.18. [95]. The zwitterionic form of NCMCS allows its binding to cells or viruses, as well as its permeability through cells. Negatively charged, ampholytic NCMCS binds to HIV-1 gp120 [95]. HIV gp120 is an important viral coat glycoprotein receptor for HIV-1 entry into host cells [122]. By attaching to gp120, NCMCS stops HIV-1 specificity to adhere to specific CD4+ cell receptors [95]. NCMCS inhibits competitively the virus-specific reverse transcriptase [122]. NCMCS is expected to limit HIV-1 infection by preventing virus-cell membrane contact to hinder the processes of reverse transcription of the viral genome. Deoxy-6-bromo-N-phthaloyl chitosan (6BrNPC) possesses antiviral activity against NDV [123]. It has a molecular weight of 1160 kDa and a bromine substitution degree of 0.5471. Their activity is higher than non-brominated variants of chitosan (N-phthaloyl chitosan and 6-aminoethylamino-6-deoxy-Nphthaloyl chitosan). 6BrNPC elevates TNF- and IFN- in hosts, which stimulate the immune system against NDV and impede virus transcription [123]. Chitosan-sialyllactose compound mediates with influenza viruses through its sialyllactose component’s high affinity binding with the virus’s hemagglutinin; this binding limits viral attachment to host cells to trigger infection [124]. The conjugate of chitosan-peptide bind to the viral glycoprotein gp41, preventing it from connecting with the CD4 host cell membrane receptors for HIV-1 fusion alike with infected and uninfected cells. Poly-quaternary phosphonium oligochitosan (PQPOC) with 0.2855 degrees of poly-quaternary phosphonium substitution is more effective against feline calicivirus, hepatitis A virus, and Coxsackievirus B4 than that with a lower substituent content [125]. PQPOC has a greater ionic character than zwitterionic chitosan because of its persistently positive moiety. The electro-positive nature of PQPOC is anticipated to attract negatively charged viral capsids, resulting in structural damage to the viral capsids and so inhibiting viral reproduction. The chitosan backbone of PQPOC may suppress virus spread by modulating ribonuclease express in hosts, hence enhancing viral RNA structural breakdown [126]. Chitosan sulphate derivatives inhibit HIV-1 (human immunodeficiency virus-1) replication in a specific manner. It inhibits entry of viruses into T cells by blocking the mediation between viral gp120 and the CD4 receptor in the culture. By competitive inhibition, it also prevents reverse transcriptase from attaching to poly-A [95]. Anionic chitosan derivatives electrostatically are attracted towards positively charged gp120 molecule and inhibit fusion of virus with cell membranes. It is reported that the sulphate groups on glucosamine residues and their position are responsible for the inhibitory action of chitosan [93]. Ishihara et al. employed sulfated derivatives of chitin and chitosan microparticles in vitro and in vivo to combat herpes simplex virus type-1 (HSV-1) and Friend murine leukemia assist virus [109]. According to this group’s research, these chemicals reduce the virus’s pathogenicity by interfering with its absorption and penetration into cells [108]. Investigation of antiviral properties of silver nanoparticle/chitosan composites in relation to the H1N1 influenza virus are also available [101]. In this study chitosan was made into a polymer matrix to immobilize silver particles, restrict their dispersion in the environment, and decrease nanoparticle toxicity. The smaller the size of silver particle, the better is its antiviral ability [102]. Hassan et al. [109] demonstrated that chitosan nanoparticles protect Vero cells from the cytopathic effects of Rift Valley fever (RVF), coxsackie viruses, and HSV-1 by inhibiting the replication of these viruses. Chitosan nanoparticles may also be employed to deliver drugs to target viruses and improve their therapeutic potency and reduce dosage. Noroviruses are enteric viruses that are of concern as they cause gastroenteritis in humans. Morphology-wise they are single stranded, envelope-less, mostly positive-sense RNA viruses. Chitosans of varied molecular weights have shown potential against cultivable norovirus surrogates and lowered viral titers [127]. With regard to direct effect on antiviral activity, it is widely recognized that chitin and chitosan degrade the structural integrity of viruses via surface charge interactions. Positively charged chitosan attracts towards negatively charged viral capsid proteins and consequently dismantles the viral structure. In addition, chitosan compounds tend to bind with tail fibers of virus and cause structural dissociation, as well as prohibit viral replication [76,128].

Figure 3, gives the list of viruses that have been inhibited using chitosan composites/derivatives.

## 4. Anti-COVID-19 Applications of Chitosan Composites and Derivatives

During the pandemic, mutation strains of SARS-CoV-2/COVID-19 [129,130,131,132] have enhanced the risk of re-infection and undermine already built immunity and vaccination efficacy [133,134]. Alternately, vaccinations that induce broadly neutralizing antibodies could be developed [135,136]. About 23 COVID-19 vaccines were licensed under emergency conditions in humans by 2021, and the rest (about 329) were in clinical (111) or preclinical (218) stages [137]. These included inactivated whole viruses (such as CoronaVac and Covaxin), mRNA-loaded liposomes (such as BNT162b2 and mRNA-1273), adenovirus vectors (such as ChAdOx1 nCoV-19, CTII-nCoV, and Sputnik V), and virus-like particles (such as NVX-CoV2373) [138]. In phase 3 investigations, these vaccines demonstrated a 65–96% antiviral efficacy against illness and mortality [139,140,141,142,143]. The vaccinations work because the S protein extends from the virus’s surface and is recognized by angiotensin-converting enzyme 2, which enhances viral uptake [144]. Nonetheless, the specificity of vaccines towards S protein declines as a consequence of the rapid accumulation of mutations [145,146,147,148]. Mutations include L18F, D80A, D215G, and 242–244 in the N-terminal domain; K417N, E484K, and N501Y in the receptor-binding domain (RBD); and D614G and P681R in other places that sustain spike stability and function [149,150,151].

In light of the increasing incidence and mortality rates of COVID-19 infection worldwide, the development of treatment techniques has attracted the highest level of interest. The antiviral potential of marine resources is underutilized. Chitosan (polycarbohydrate) is a prevalent glycan that is bioactive and abundant in marine species. The structural availability of reactive amine/hydroxyl groups invoking minimum toxicity/allergenicity has caught attention to study the anti-SARS-CoV-2 activity of chitosan compound. Due to the strong docking affinity of the ligands (−6.0 to −6.6 kcal/mol) with SARS-CoV-2 spike protein RBD, N-benzyl-O-acetyl-chitosan, imino-chitosan, and sulfated chitosan oligosaccharides derivatives are ranked as promising antiviral candidates. In fact the results of the ADMET experiment are extremely favoring to conduct more clinical trials of the above mentioned chitosan compounds as an agent against SARS-CoV-2 treatment [152].

Milewska et al. created chitosan-based anti-coronavirus compounds that prevent in vitro and in vivo infection [113]. Indeed, investigation of the interaction between this chitosan derivative and the recombinant ectodomain of the S protein showed the development of polymer–protein complexes as a result of binding. They showed that the polymer inhibits the interaction between the spike protein of the coronavirus and the cellular receptor [113]. In 2020, the above group simulated cell culture models to simulate the coronavirus-replicating layer lining conductive airways. The model culture was loaded with virus and exposed to the chitosan compounds. The model investigation showed that the polymeric system successfully suppressed SARS-CoV-2 and MERS-CoV [153]. The combination of chitosan and DNA found to greatly reduce lung inflammation and boosts host antiviral mechanism. The intranasal delivery of a DNA vaccine containing SARS-CoV nucleocapsid protein and chitosan nanoparticles to present antigens to T cells by nasal-resident DCs has been tested [154]. This approach activates systemic IgG and nasal IgA antibodies targeting N protein of the virus, hence enhancing the immune response against SARS-CoV.

Milewska et al. studied the SARS-CoV-2 inhibitory factor of N-(2-hydroxypropyl)-3-trimethylammonium chitosan chloride (HTCC) and its hydrophobic derivative [113]. HCoV-NL63 and human murine virus were suppressed by these polymers. Extensive investigation into the pathways indicated that the chitosan polymer formed a compound with the ectodomain of the HCoV-spike NL63 protein without producing considerable cytotoxicity. In addition, it has been suggested that inhibition is entirely triggered by site-specific binding [155]. A later study on viral entry suppression by HTCC polymeric compound in the LLC-MK2 cell line revealed intriguing results of polymer disrupting the receptor cell signaling with a spike protein via aggregation of spike [114]. Additionally, it eliminates the colocalization of ACE-2 receptors with viruses. The spectrum of suppressed human coronavirus (HCoV-OC43, HCoV-229E, HCoV-NL63, and HCoV-HKU1) infections varied according to the degree of substitution. Later, the efficacy of HTCC was tested in pathogenic coronaviruses, MERS-CoV, and SARS-CoV-2 using Vero and Vero E6 cell lines and the human airway epithelial (HAE) model. The study report revealed that HTCC has the ability to suppress viral reproduction especially on the SARS-CoV-2 virus. Moreover, electrostatic attraction with viral spike protein significantly restricted the uptake of viruses into susceptible cells [156].

Even after recovery, COVID-19-caused respiratory disease is currently a major worldwide health concern. Since infected patients’ lung lesions are still characterized by acute respiratory distress syndrome, including alveolar septal edema, pneumonia, hyperplasia, and hyaline membranes, it is vital to uncover other options that can overcome the inflammatory process and improve COVID-19 treatments. The polyphenolic extracts were integrated into bovine serum albumin (BSA) molecules, which were then coated with chitosan, a polymer that promotes mucoadhesion [157].

The efficiency of the silver/chitosan coating system is boosted in combination with adhering curcumin from *Curcuma longa* rhizome extract, resulting in nanosilver particles. *Curcuma longa* rhizome aids in the nanobiotechnological manufacture of silver/chitosan nanocomposite antibacterial surface coating. Antiviral drugs including nanosilver and curcumin have a synergistic effect that minimizes contamination. A simple coating technology is suited to the maximum number of exposed contact surfaces with metallic, ceramic, polymeric, or wooden surfaces, etc. Handles, rails, switches, touch displays, keyboards, tabletops, etc. than self-sterilize and function normally. They are implementing preventative measures against the COVID-19 infection linked to SARS-CoV-2 [158].

This work includes systematic research of the synthesis of chitosan hydrogels with dicarboxylic acids (malic and glutaric acid) and their detailed characterization (Fourier transform infrared spectroscopy, determination of cross-linking efficiency, rheological studies, thermal analysis, and swelling kinetics). Without additives or catalysts, chitosan hydrogels can be chemically cross-linked using malic or glutaric acid, as demonstrated by the results. In addition, the ability of hydrogels to adsorb three separate ACE2 inhibitors as active medicinal components has been investigated. The API content and mucoadhesive property of hydrogels can provide a suitable foundation for the development of a nasal formulation to reduce the risk of SARS-CoV 2 infection [136,159]. Extensive literature reviews on the biocompatibility of chitosan nanoparticles allow this experiment to be conducted quickly on humans. Moreover, it is easy to link antibodies to chitosan, hence we chose these nanoparticles [160].

Using chitosan/polycaprolactone bioink, the primary purpose of this study was to construct a biocompatible and mechanically sound 3D-printed scaffold for lung tissue engineering. Several 3D printing compositions were analyzed using the Design-Expert program. The selected scaffolds were tested for their ability to support MRC-5 cell growth, proliferation, and migration. Based on the results, the average diameter of the chitosan/polycaprolactone filaments was determined to be 360 mm. The printability was determined by the chitosan concentration, whereas polycaprolactone content had no effect. Despite the fact that the scaffolds’ polycaprolactone content may be modified, they exhibited promising swelling, degradation, and mechanical properties [161].

To determine a specific treatment for COVID-19, silymarin–chitosan nanoparticles (Sil–CNPs) were evaluated for their potential as antiviral agents against SARS-CoV-2 using in silico and in vitro techniques. Sil and CNPs were docked with SARS-CoV-2 spike protein using AutoDock Vina. Both Vero and Vero E6 cell lines were tested for cytotoxicity using the MTT assay. Using crystal violet, the IC50 was determined at doses of 0.91, 12.2, and 0.80 mg/mL for CNPs, Sil, and Sil–CNPs, respectively, in virucidal/replication studies. These findings reveal the antiviral activity of Sil–CNPs against SARS [162].

Electrospinning the fibers directly on the surface of the textile or fabric to create a composite fabric is an easy way for incorporating electrospun fibers into textile. This process of putting fibers on fabric is inexpensive and reduces manufacturing procedures, making it suited for mass production. In addition, electrospun chitosan nanofibers were effectively mass-produced, and their large-scale production with the Force spinning method was previously described [163].

The cytokine storm syndrome (CSS) leads to high patient mortality, especially those who have preconditions like ALI (acute lung infection) and ARDS (acute respiratory distress syndrome) resulting from COVID-19. Currently, no standard treatment model exists for ALI- or ALI/ARDS-triggered CSS. Consequently, the development of potent drugs and therapeutic protocols for CSS associated with ALI/ARDS is the need of the hour. Due to the targeted capability of drug delivery to organs like lungs via nasal inhalation, chitosan compounds holds promises for treating inflammatory lung diseases. Enhancing the absorption and bioavailability of poor water-soluble medications to therapeutic level in the nasal-mucosal region offers more promises in the treatment of ALI/ARDS. For example, anti-inflammatory hesperidin-loaded chitosan nanoparticles (HPD/NPs) were nasally administered to inflamed lungs. In an inflammatory context, HPD/NPs demonstrated higher cellular absorption in vitro and in vivo compared to HPD alone. In a mouse model, HPD/NPs greatly reduced lung damage by decreasing the levels of pro-inflammatory cytokines and decreased vascular permeability [164].

As a result of its better nanoscale characteristics, nanocomposites have garnered considerable interest in a variety of applications. Sol-gel technology was utilized to synthesize a chitosan–zeolite–ZnO nanocomposite, and a molecular docking study was conducted to assess its antiviral efficacy against SARS-CoV-2 [165].

Sun et al. (2009) manufactured biotinylated chitosan nanoparticles encapsulated with bovine coronavirus N-protein [166]. They examined the effectiveness of their vaccine against SARS-CoV-2. Researchers employed a bifunctional fusion protein vector to selectively target dendritic cells. The increase in mucosal IgA and systemic IgG against N-protein suggests that chitosan is a viable carrier for gene transfer [166,167]. Plasmid DNA encoding the nucleocapsid of SARS-CoV-2 was delivered intranasally through chitosan nanoparticles. This stimulates the release of SARS-CoV-2 spike protein, which competes with live coronavirus for ACE2 receptor binding. The results suggested that mice produce significant quantities of IgA and IgG. Chitosan reduces the breakdown of DNA vaccines, and its cationic nature enhances DNA binding. In addition, the mucoadhesive characteristics of chitosan stimulate its usage as a mucosal delivery adjuvant [168,169,170]. Table 1 summarizes the anti-COVID-19 applications of chitosan composites/derivatives.

## 5. Future Perspective and Recommendations

Figure 3 outlines the various facets of Anti-COVID-19 challenges that chitosan derivatives have been demonstrated to address. We have reviewed the various pioneering antiviral applications of chitosan and its derivatives, specifically anti-COVID-19 applications. Their antiviral properties depended on molecular weight, deacetylation, chitosan concentration, and ambient pH. This review shows that chitosan can be used to make broad-spectrum antiviral drugs. Chitosan’s antiviral action is improved by conjugating with hydroxypropyl trimethylammonium, sulphate, carboxymethyl, bromine, sialyllactose, peptide, and phosphonium moieties. Chitosan conjugates including HTCC, sulfated chitosan, SCM-chitin, NCMCS, 6BrNPC, and SCC have broad-spectrum antiviral action against human coronavirus, HIV, herpes simplex virus, influenza virus, NDV, human papillomavirus, and F-MuLV. Their antiviral activity relies on molecular weight, acetylation, conjugate substitution, and location. Chitosan monomeric residues have substituents. For nanomedical purposes, resultant conjugates have been mixed with silver nanoparticles and other antiviral drugs [125].

At effective in vitro therapeutic dosages, chitosan and its derivatives have negligible cytotoxicity. Chitosan and its derivatives are being studied as antiviral treatments in vitro. However, cell culture studies may not fully account for their antiviral actions in organs and complex systems within animals. To understand chitosan-based therapeutics’ potential and side effects, in vivo testing is needed.

Chitosan may inhibit SARS-CoV-2 by attaching to the spike glycoprotein trimer cavity [171]. SARS-CoV-2, an enveloped virus, can be treated with chitosan-based treatments that take into account the structure-activity relationship of chitosan derivatives substituted with carboxymethyl, sulphate, bromine, N-(2-hydroxypropyl)-3-trimethylammonium, phthaloyl, and sialyl-lactose moieties that are active against other enveloped viral variations. SARS-CoV-2-specific optimization studies must optimize molecular weight, degree of polymerization, acetylation, degree, and site of conjugate substitution of chitosan that will provide the greatest antiviral effect against SARS-CoV-2. Further, conjugating chitosan with optimal antiviral characteristics by other anti-SARS-CoV-2 ligands may confer additive or synergistic effects and greater effectiveness in inhibiting SARS-CoV-2 replication.

Pandemic preparations include accelerating vaccine research and reusing approved antivirals and other medications [175]. COVID-19 may spread too quickly with such a method. Since the medicine may not function against a mutation, developing an antiviral treatment for a single virus may not be worthwhile [175]. The Omicron version SARS-CoV-2 mutated quickly. Thus, a broad-spectrum antiviral technique will be the next-generation remedy for COVID-19 and other new viruses. This strategy could enable viable treatments to dramatically improve pandemic preparedness in early outbreaks. These antiviral chitosan compounds are broad-spectrum. They may create novel anti-SARS-CoV-2 therapies in the future. Currently, only a few laboratory experiments support the broad-spectrum efficacy of chitosan derivatives. More validated and optimized studies, supported by clinical trials, are needed.

Chitosan-based nanomedicines have various advantages that potentially give target-specific and cost-effective treatments for respiratory disorders such asthma, COPD, lung cancer, pneumonia, COVID-19, and others. Chitosan-based nanomedicines can be made in solid, liquid, or gel form and directly affect lungs. Nanomedicines can efficiently enter the respiratory system, target the lungs, and continually release API from the vehicles, making them vital in treating respiratory illnesses. Despite its benefits, few nanomedicines have been employed clinically for daily usage. Chitosan derivative-based nanomedicines could boost antiviral capabilities, another topic for research. Value-added chitosan derivative medicinal compounds are few. Chitosan is a stable and effective pharmacological excipient because of its solubility, stability, gene transport to the lungs, and controlled release of medicines, proteins, and peptides. This natural resource has plenty to be utilized. This review encourages researchers to expand chitosan derivatives and composites for anti-COVID-19 uses and pharmaceutical formulations.

## 6. Conclusions

The various antiviral activities of chitosan composites and derivatives were discussed. Their mode of action and the list of chitosan derivatives that have been proven for their antiviral effects were listed. The anti-COVID 19 potential of chitosan composites and derivatives was summarized. The gaps in the existing knowledge and the recommendations for future direction have been put forth.

## Figures and Tables

**Figure 1 antibiotics-12-00665-f001:**
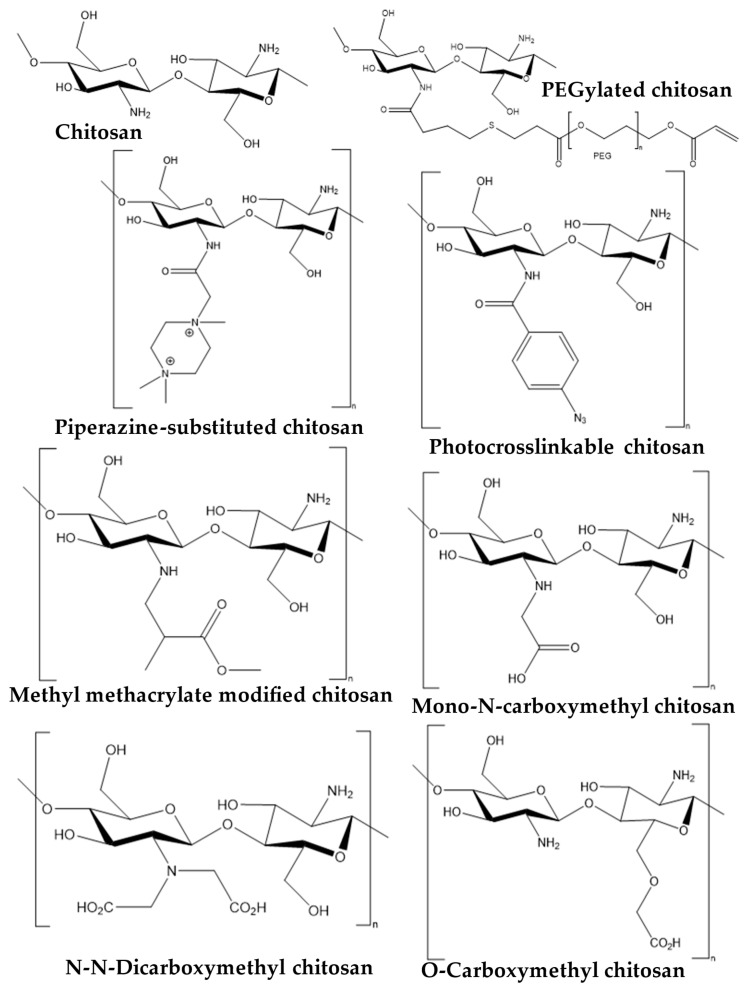
Chitosan derivatives that have been confirmed for their antiviral activity.

**Figure 2 antibiotics-12-00665-f002:**
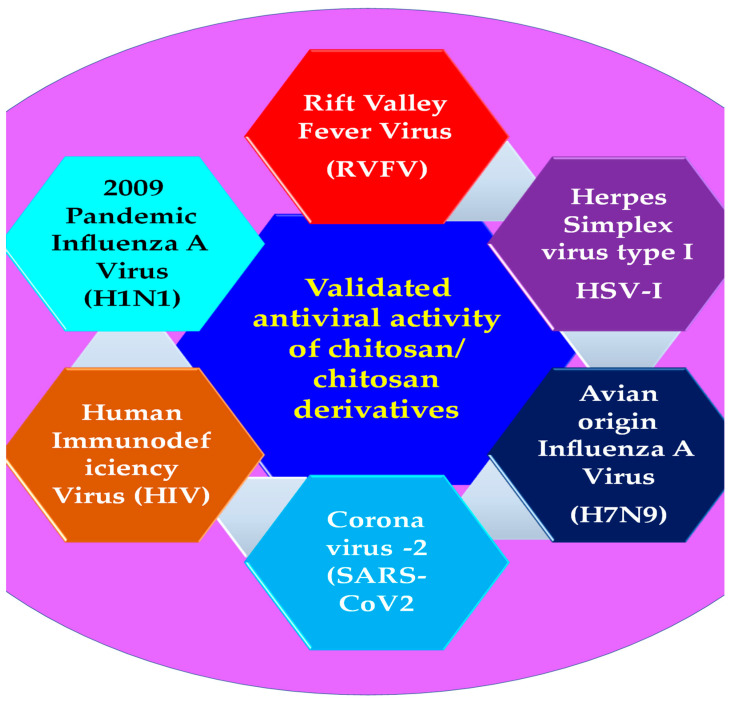
Overview of the various viruses that are susceptible to the antiviral activity of chitosan derivatives.

**Figure 3 antibiotics-12-00665-f003:**
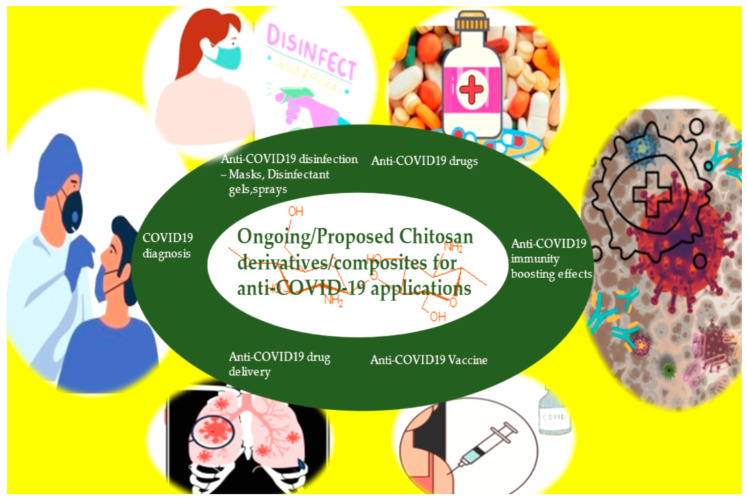
Contributions of chitosan derivatives to combat various anti-COVID-19 challenges.

**Table 1 antibiotics-12-00665-t001:** Anti-COVID 19 applications of chitosan composites/derivatives.

Chitosan Composite/Derivative	Anti-COVID Application	Antiviral Mechanism	References
Chitosan nanoparticle in aerosol formulations for anti-COVID-19 drugs (Novochizol™)	Chitosan NPs in Novochizol™ adhere to mucosal membranes and deliver drugs, siRNA, and peptides	Slow release of anti-COVID-19 drug in the infected area by slow degradation of chitosan	[76]
Chitosan and genipin nano/micro spheres	Antiviral activity against HCoV-NL63 and HCoV-OC43	Sphere complex exerts electrostatic interaction with Viral nucleic acid	[112]
A cationically modified chitosan derivative, N-(2-hydroxypropyl)-3-trimethylammonium chitosan chloride (HTCC), and hydrophobically-modified HTCC	Chitosan derivatives as inhibitors for coronavirus NL63	Act as inhibitors of HCoV-NL63 replication	[113]
Cationic modifications of chitosan, N-(2-hydroxypropyl)-3-trimethylammonium chitosan chloride (HTCC), and hydrophobically-modified derivative (HM-HTCC)	Inhibition of the coronavirus HCoV-NL63	HTCC and HM-HTCC inhibit interaction of a virus with its receptor	[114]
Chitosan hydrogels with dicarboxylic acids (malic and glutaric acid)	Viral spike protein to the ACE2 receptors	Controlling the risk of SARS-CoV-2 infection.	[159]
Chitosan nanoparticles (CNPs)	CNPs encapsulated with silymarin (Sil–CNPs)	Mechanism was studied	[161]
Chitosan nanofibers made from N,N,N-trimethyl chitosan (TMC), the single N-quaternized (QCS) and the double N-diquaternized (DQCS)	Making Personal Protective Equipment (PPE) with chitosan fibers	SARS-CoV-2 viral repulsion by Cationic amino group	[163]
Zero-valent nanosilver/titania-chitosan (nano-Ag0/TiO2-CS) filter bed	Removal of viral aerosols to minimize the risk of airborne transmission	Reducing infection caused by COVID-19 tested using MS2 bacteriophages	[164]
Chitosan-Zeolite-ZnO nanocomposite	Identification of nanocomposite and SARS-CoV-2 glycoprotein receptors RCSB and PDB interaction by molecular docking	Antiviral activity chitosan–zeolite and ZnO against SARS-CoV-2 The SARS-CoV-2 ligand and receptor	[165]
A cationically modified chitosan derivative, HTCC	Chitosan derivative HTCC as inhibitors for coronavirus MERS-CoV and SARS-CoV-2 infection	Inhibiting receptor protein	[153]
Chitosan nanoparticles	Chitosan nanoparticles act as carrier molecules for DNA vaccine expressing SARS-CoV nucleocapsid protein	Activating T cells to produce IgG and nasal IgA antibodies against SARS-CoV	[154]
Chitosan and its derivatives (In silico studies)	Inhibition of SARS-CoV-2 viral binding with host cells	Inhibition of viral spike protein binding to the ACE2 receptors	[171]
β-chitosan	Inhibition of SARS-CoV-2 viral binding with host cells	Binding interaction between SARS-CoV-2 S-RBD and ACE2 receptors	[172]
Nanochitosan	Application as vaccine adjuvant or vehicles for delivery of	Vaccine delivery via the intranasal route to fight against SARS-CoV-2	[173]
Amphiphilic chitosan Npalmitoyl-N-monomethyl-N,N-dimethyl-N,N,N-trimethyl-6-O-glycolchitosan (GCPQ),	As a COVID-19 prophylactic agent	Reduce the infectivity of SARS-CoV-2 in A549ACE2+ and Vero E6 cells and in human airway epithelial cells	[174]

## Data Availability

Not applicable.

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
