# Peer review of "Anti-COVID-19 Credentials of Chitosan Composites and Derivatives: Future Scope?"

_antibiotics, 2023, doi:10.3390/antibiotics12040665_

Round 1

Reviewer 1 Report

Comments: The authors have compiled the review on the utility of chitosan composites and derivatives for anti-COVID-19 activity. The review is broadly divided into 4 sections.

The section 3 can be sub-sectioned by mode of action, type of chitosan etc. Similarly for the section 4. It is very heavy reading when all the information is to be read consecutively.

English language requires editing.

The review may be accepted after major revisions.

Specific comments

Line 42: chitosan monograph was added to the US National Formulary in 2008 42 and the European Pharmacopoeia in 2011. Add the necessary reference.

Line 46: Give the names of countries producing the products

Line 49: Give the name of country producing for Anscare.

Line 50 : rewrite the sentence

Other than blood clotting, chitosan has nu-50 merous biomedical applications such as anti-bacterial [13–15]; gene delivery [16]; bio-51 composites [17–20]; nanoengineering [21–23] and excipients [9,24].

Line 54: rewrite the sentence for clarity

By optimizing the wettability, the hydrophobic part of chitosan pene-54 trates the lipid bilayer of the bacterium and affects the integrity of cell wall.

Line 83: while bactericidal potential of chitosan is increased through 83 PEGylation [35,36].

Line 86: Moreover, the presence of methyl 85 methacrylate moiety in chitosan has shown to be useful for drug delivery application [38].

Line 110: Differentiate between drugs and steroids

The current drugs used to treat 110 COVID-19 are Remdesivir, Enoxaparin, Methyl Prednisolone, Dexamethasone, Tocili-111 zumab and Ivermectin.

Line 134: Apart from textiles, chitosan nanopar-134 ticle loaded VESTA respirators were developed against pathogens. Add the necessary reference.

Besides these, a clini-135 cal study (NCT04490200) involving N95 mask with and without chitosan is underway on 136 1000 clinical professionals to check for its effectiveness against COVID-19 infection, that 137 whether the chitosan nanoparticles can adsorb and inactivate the virus when they come 138 in contact with it. Add the necessary reference.

Line 139: rewrite the sentence for clarity

120 patients at Jinan Military General Hospital were enrolled in a clinical 139 trial utilizing anti-microbial chitosan dressing gel to treat nonneoplastic epithelial diseases 140 involving the skin and vulvar mucosa (NCT02890277)[57]

Line 149:   in vivo – in italics

rewrite the sentence for clarity – chitosan against bacteria infected mice

Line 153: Explain the abbreviations when mentioned for first time MSSA

Lin1 184: Additionally, chitosan and fluconazole worked well to treat drug-184 resistant bacterial strains [67].

Lo, W.-H.; Deng, F.-S.; Chang, C.-J.; Lin, C.-H. Synergistic Antifungal Activity of Chitosan with Fluconazole against Candida 859 Albicans, Candida Tropicalis, and Fluconazole-Resistant Strains. Molecules 2020, 25, doi:10.3390/molecules25215114.

Fluconazole is an antifungal drug, its activity against bacteria

Please reread your article carefully for the statements

Line 191: Candida, Pneumocystis, Aspergillus and Cryptococcus [68].

Organism genus names in italics line 191 , 202, 204 and elsewhere

Line 243: The EC (emulsifiable concentrate) of sulfated chitosan oligosac-243 charide was 2.19 g/ml, and it had a lytic effect (EC 1.43 g/ml)[89] against HIV-1 virus.

Artan, M.; Karadeniz, F.; Karagozlu, M.Z.; Kim, M.-M.; Kim, S.-K. Anti-HIV-1 Activity of Low Molecular Weight Sulfated 903 Chitooligosaccharides. Carbohydr. Res. 2010, 345, 656–662.

The following statement is from the reference:

In reference Among SCOSs, SCOS III also exhibited to be the most potent inhibitor of HIV-1 and protected the cells from lytic effect of HIV-1 with an EC50 of 1.43 μg/ml.

Please reread your article carefully for the statements

Line 249 It led to the conclusion that chitosan-conjugate 249 nanoparticles might be employed for targeting and producing long-lasting polymeric pro-250 drugs with increased therapeutic potential and fewer unwanted effects for antiretroviral 251 therapy [90]. Loutfy et al. fabricated curcumin loaded chitosan (low molar mass) nano-252 particles to boost anti-retro viral potency [91].

90. Loutfy, S.A.; Elberry, M.H.; Farroh, K.Y.; Mohamed, H.T.; Mohamed, A.A.; Mohamed, E.B.; Faraag, A.H.I.; Mousa, S.A. 909 Antiviral Activity of Chitosan Nanoparticles Encapsulating Curcumin Against Hepatitis C Virus Genotype 4a in Human 910 Hepatoma Cell Lines. Int. J. Nanomedicine 2020, 15, 2699–2715.

Article by Loutfy eta al 2020 is retracted from the journal

Please do not quote retracted articles

Numbering of the reference is not correct.

Line 258: Moloney murine leukaemia virus with Heparan sulphate receptor investigation on 258 SC-1 and NIH-3T3 cell lines concluded that sulphate replacement on C4H6O4 of succinyl chitosan structure exhibited reasonable inhibitory effect out of dextran sulphate chitosan  matrices as a positive control.

Reframe the sentence.

Line 404: Sulfated 402 6-O-carboxymethyl-chitin (SCM-chitin) (430 kDa, 0.0766-1.69 degree of sulphate substitu-403 tion, 0.56 degree of carboxymethyl substitution) greatly suppresses Friend.

Correct As Friend murine leukaemia here and elsewhere in the manuscript

Line 264: In addition, the addition of sulfonic groups to the hydroxyl 264 group of C-6 did not have a significant effect on anti-HIV-1 efficacy. They hypothesised that the anti-HIV-1 activity of sulfonic groups at C-2 and C-3 is due to anticoagulant interactions with the GP120 envelope glycoprotein, whereas replacement at C-6 had non-specific ionic interactions with blood coagulation proteins [94].

Nishimura, S.I.; Kai, H.; Shinada, K.; Yoshida, T.; Tokura, S.; Kurita, K.; Nakashima, H.; Yamamoto, N.; Uryu, T. Regioselective 917 Syntheses of Sulfated Polysaccharides: Specific Anti-HIV-1 Activity of Novel Chitin Sulfates. Carbohydr. Res. 1998, 306, 427–433.

Abstract of paper: The regioselective introduction of sulfate group(s) at O-2 and/or O-3 had little effect on generating anticoagulant activity, whereas 6-O-sulfated chitin strongly inhibits blood coagulation. These results suggest that the specific interaction of these new types of chitin sulfates with gp 120 of the AIDS virus depends significantly on the sites of sulfation rather than on the total degree of substitution on sugar residues.

The message in Line 264 is not as mentioned in the quoted reference of Nishimura et al 1998.

Line 286: It also emphasised site selectivity, citing the improved anti-HIV-1 efficacy of pol-286 ysaccharides sulfated at Oxygen-2 and/or Oxygen-3 sites [97].

Correct as O-2 and/or O-3 sites

Line 335:Paul T discovered the antiviral effect of chitosan on an adenovirus that causes 335 pulmonary issues in young children over the course of his scientific research.

Line 360 In vitro in italics and elsewhere

Line 368: It is reported the efficiency of viral load removal gets improvised after sul-368 phate substitution.

Not clear, rewrite the sentence

Line 467: We employed site-specific docking to analyse a library of chitosan derivatives at the spike protein 468 Receptor Binding Domain (RBD) of SARS-CoV-2 wild type, B.1.1.7 (UK), and P.1 (Brazil).

Is this the author’s work if not reframe the sentence.

Elsewhere in manuscript please make suitable ocrrections

Line 528: Besides curcumin, nano-528 composites too found to impede virus entry and multiplication in Huh7 cells [91].

Reframe the sentence

Author Response

Reviewer 1

Comments: The authors have compiled the review on the utility of chitosan composites and derivatives for anti-COVID-19 activity. The review is broadly divided into 4 sections.

The section 3 can be sub-sectioned by mode of action, type of chitosan etc. Similarly for the section 4. It is very heavy reading when all the information is to be read consecutively.

English language requires editing.

The review may be accepted after major revisions.

Ans. We would like to thank the Reviewer for the valuable and extensive specific comments. Your suggestions have been instrumental in improving the quality of our manuscript, Thank you for your time and efforts. Thank you.

Specific comments  

Line 42: chitosan monograph was added to the US National Formulary in 2008 42 and the European Pharmacopoeia in 2011. Add the necessary reference.

Ans. Added

Line 46: Give the  of countries producing the products

Ans. Information added in revision. Thank you.

These products are used in many countries. This info has been added to the manuscript.

Axiostat®- is FDA approved and in use in 40 plus countries (https://axiobio.com/about-axiobio/)

ChitoFlex®- https://tricolbiomedical.com/about-tricol/

CeloxTM - https://www.celoxmedical.com/instructions-for-use-ifu/

Line 49: Give the name of country producing for Anscare.

Ans. Taiwan

Line 50 : rewrite the sentence

Ans. Amended and thanks.

Other than blood clotting, chitosan has numerous biomedical applications such as anti-bacterial [13–15]; gene delivery [16]; bio-51 composites [17–20]; nanoengineering [21–23] and excipients [9,24].

Ans. Sentence rephrased. Thanks

Line 54: rewrite the sentence for clarity

Ans. Sentence 53-55 amended. Thanks

By optimizing the wettability, the hydrophobic part of chitosan pene-54 trates the lipid bilayer of the bacterium and affects the integrity of cell wall.

Ans. Sentence rephrased. Thanks

Line 83: while bactericidal potential of chitosan is increased through 83 PEGylation [35,36].

Ans. Sentence rephrased. Thanks

Line 86: Moreover, the presence of methyl 85 methacrylate moiety in chitosan has shown to be useful for drug delivery application [38].

Ans. Sentence rephrased. Thanks

Line 110: Differentiate between drugs and steroids

Ans. We haven’t addressed steroids anywhere, so we restrict to the given information. Thanks

The current drugs used to treat 110 COVID-19 are Remdesivir, Enoxaparin, Methyl Prednisolone, Dexamethasone, Tocili-111 zumab and Ivermectin.

Ans. Sentence modified with expected info. Thanks

Line 134: Apart from textiles, chitosan nanopar-134 ticle loaded VESTA respirators were developed against pathogens. Add the necessary reference.

Ans. Added. Thanks

Besides these, a clini-135 cal study (NCT04490200) involving N95 mask with and without chitosan is underway on 136 1000 clinical professionals to check for its effectiveness against COVID-19 infection, that 137 whether the chitosan nanoparticles can adsorb and inactivate the virus when they come 138 in contact with it. Add the necessary reference.

Ans. Added. Thanks

Line 139: rewrite the sentence for clarity

Ans. Sentence modified

120 patients at Jinan Military General Hospital were enrolled in a clinical 139 trial utilizing anti-microbial chitosan dressing gel to treat nonneoplastic epithelial diseases 140 involving the skin and vulvar mucosa (NCT02890277)[57]

Ans. Sentence modified

Line 149:   in vivo – in italics

Ans. Modified. Thanks

rewrite the sentence for clarity – chitosan against bacteria infected mice

Ans. Accordingly re-written. Thanks 

Line 153: Explain the abbreviations when mentioned for first time MSSA

Ans. Abbreviated. Thanks

Lin1 184: Additionally, chitosan and fluconazole worked well to treat drug-184 resistant bacterial strains [67].

Ans. Sentence modified

Lo, W.-H.; Deng, F.-S.; Chang, C.-J.; Lin, C.-H. Synergistic Antifungal Activity of Chitosan with Fluconazole against Candida 859 Albicans, Candida Tropicalis, and Fluconazole-Resistant Strains. Molecules 202025, doi:10.3390/molecules25215114.

Fluconazole is an antifungal drug, its activity against bacteria

Please reread your article carefully for the statements

Ans. Sorry about that. Corrected.

Line 191: Candida, Pneumocystis, Aspergillus and Cryptococcus [68].

Organism genus names in italics line 191 , 202, 204 and elsewhere

Ans. Accordingly corrected. Thanks.

 Line 243: The EC (emulsifiable concentrate) of sulfated chitosan oligosac-243 charide was 2.19 g/ml, and it had a lytic effect (EC 1.43 g/ml)[89] against HIV-1 virus.

Artan, M.; Karadeniz, F.; Karagozlu, M.Z.; Kim, M.-M.; Kim, S.-K. Anti-HIV-1 Activity of Low Molecular Weight Sulfated 903 Chitooligosaccharides. Carbohydr. Res. 2010345, 656–662.

The following statement is from the reference:

In reference Among SCOSs, SCOS III also exhibited to be the most potent inhibitor of HIV-1 and protected the cells from lytic effect of HIV-1 with an EC50 of 1.43 μg/ml.

 Ans. Sorry, reference numbering had an issue. Accordingly corrected. Thanks.

Please reread your article carefully for the statements

Line 249 It led to the conclusion that chitosan-conjugate 249 nanoparticles might be employed for targeting and producing long-lasting polymeric pro-250 drugs with increased therapeutic potential and fewer unwanted effects for antiretroviral 251 therapy [90]. Loutfy et al. fabricated curcumin loaded chitosan (low molar mass) nano-252 particles to boost anti-retro viral potency [91].

  1. Loutfy, S.A.; Elberry, M.H.; Farroh, K.Y.; Mohamed, H.T.; Mohamed, A.A.; Mohamed, E.B.; Faraag, A.H.I.; Mousa, S.A. 909 Antiviral Activity of Chitosan Nanoparticles Encapsulating Curcumin Against Hepatitis C Virus Genotype 4a in Human 910 Hepatoma Cell Lines. Int. J. Nanomedicine 202015, 2699–2715.

Article by Loutfy eta al 2020 is retracted from the journal

Please do not quote retracted articles

Numbering of the reference is not correct.

 Ans. Sorry we have removed that reference. Also checked reference numbering. Thank you.

Line 258: Moloney murine leukaemia virus with Heparan sulphate receptor investigation on 258 SC-1 and NIH-3T3 cell lines concluded that sulphate replacement on C4H6O4 of succinyl chitosan structure exhibited reasonable inhibitory effect out of dextran sulphate chitosan  matrices as a positive control.

 Ans. Changed.

Line 404: Sulfated 402 6-O-carboxymethyl-chitin (SCM-chitin) (430 kDa, 0.0766-1.69 degree of sulphate substitu-403 tion, 0.56 degree of carboxymethyl substitution) greatly suppresses Friend.

Correct As Friend murine leukaemia here and elsewhere in the manuscript

Ans. Corrected

 Line 264: In addition, the addition of sulfonic groups to the hydroxyl 264 group of C-6 did not have a significant effect on anti-HIV-1 efficacy. They hypothesised that the anti-HIV-1 activity of sulfonic groups at C-2 and C-3 is due to anticoagulant interactions with the GP120 envelope glycoprotein, whereas replacement at C-6 had non-specific ionic interactions with blood coagulation proteins [94].

Nishimura, S.I.; Kai, H.; Shimada, K.; Yoshida, T.; Tokura, S.; Kurita, K.; Nakashima, H.; Yamamoto, N.; Uryu, T. Regioselective 917 Syntheses of Sulfated Polysaccharides: Specific Anti-HIV-1 Activity of Novel Chitin Sulfates. Carbohydr. Res. 1998306, 427–433.

Abstract of paper: The regioselective introduction of sulfate group(s) at O-2 and/or O-3 had little effect on generating anticoagulant activity, whereas 6-O-sulfated chitin strongly inhibits blood coagulation. These results suggest that the specific interaction of these new types of chitin sulfates with gp 120 of the AIDS virus depends significantly on the sites of sulfation rather than on the total degree of substitution on sugar residues.

The message in Line 264 is not as mentioned in the quoted reference of Nishimura et al 1998.

 Ans. It was reference issue, we have checked on it now.

Line 286: It also emphasised site selectivity, citing the improved anti-HIV-1 efficacy of pol-286 ysaccharides sulfated at Oxygen-2 and/or Oxygen-3 sites [97].

Correct as O-2 and/or O-3 sites

Ans. Accordingly done.

Line 335:Paul T discovered the antiviral effect of chitosan on an adenovirus that causes 335 pulmonary issues in young children over the course of his scientific research.

Ans. Statement corrected and reference added. Thanks.

Line 360 In vitro in italics and elsewhere

Changed everywhere.

Line 368: It is reported the efficiency of viral load removal gets improvised after sul-368 phate substitution.

Not clear, rewrite the sentence

Statement re-written.

Line 467: We employed site-specific docking to analyse a library of chitosan derivatives at the spike protein 468 Receptor Binding Domain (RBD) of SARS-CoV-2 wild type, B.1.1.7 (UK), and P.1 (Brazil).

Is this the author’s work if not reframe the sentence.

Ans. Sentence reframed.

Line 528: Besides curcumin, nano-528 composites too found to impede virus entry and multiplication in Huh7 cells [91].

Reframe the sentence

Ans. Reframed.

Thank you so much for your valuable comments and time.

Reviewer 2 Report

The paper represents an exhaustive study regarding the use of chitosan derivatives and composites as anti-COVID 19 materials. The paper is well written and the data are clearly presented, the anti-COVID potential of chitosan derivatives being summarized and discussed in a constructive manner, the authors presenting the existing literature data but also future perspectives of the addressed problem. 

Author Response

The paper represents an exhaustive study regarding the use of chitosan derivatives and composites as anti-COVID 19 materials. The paper is well written and the data are clearly presented, the anti-COVID potential of chitosan derivatives being summarized and discussed in a constructive manner, the authors presenting the existing literature data but also future perspectives of the addressed problem.

Ans. Thank a million for your motivating and kind words. Appreciate that

Thank you.

Reviewer 3 Report

The manuscript by Gopal et al. reviewed chitosan composites and derivatives for antibacterial and antimicrobial applications with the emphasis on current understanding of the underlying mechanisms in inhibition of SARS-CoV-2. The manuscript is well organized with many facets of the antibacterial and antimicrobial properties of chitosan. A couple of minor comments are suggested to improve the quality of the manuscript:

(1) It is recommended that the authors include a table, such as table 1, to summarize the composite structures of chitosan and its derivatives.
(2) It is recommended that the authors organize section 4, with possibility to break it into sub-sections, for the reader to follow. There were anti-SARS-CoV-2 contents, HIV-1/HSV-1, H1N1 inhibition contents, and antibacterial contents. The flow is just not ideal.

Author Response

The manuscript by Gopal et al. reviewed chitosan composites and derivatives for antibacterial and antimicrobial applications with the emphasis on current understanding of the underlying mechanisms in inhibition of SARS-CoV-2. The manuscript is well organized with many facets of the antibacterial and antimicrobial properties of chitosan. A couple of minor comments are suggested to improve the quality of the manuscript:

Ans. Thank you for your encouraging words, and valuable comments. 

(1) It is recommended that the authors include a table, such as table 1, to summarize the composite structures of chitosan and its derivatives.

Ans. The composite structures have been published and summarized in numerous manuscripts, our review has already exceeded 10,000 words, so we try to restrict to the existing information. Thank you for your understanding. 
(2) It is recommended that the authors organize section 4, with possibility to break it into sub-sections, for the reader to follow. There were anti-SARS-CoV-2 contents, HIV-1/HSV-1, H1N1 inhibition contents, and antibacterial contents. The flow is just not ideal.

Ans. We terribly apologize for this mistake, actually section 4 is exclusively on  anti covid activity. But somehow we have added these out of context parts. We have now deleted the antibacterial info  and shuffled the antiviral part. Now its all done. Thank you for rightly pointing out.